A new Meckel’s cartilage from the Devonian Hangenberg black shale in Morocco and its position in chondrichthyan jaw morphospace

Greif Merle merle.greif@pim.uzh.ch 1
Ferrón Humberto G. 2
Klug Christian 1
1 Palaeontological Institute and Museum, University of Zürich , Zürich , Switzerland
2 Instituto Cavanilles de Biodiversidad i Biología Evolutiva, Universitat de València , Paterna, Valencia , Spain
Piñeiro Graciela
Electronic publication date: 2022 Dec 21
Publication date: 2022
Volume: 10
Electronic Location ID: e14418
Received 2022 Aug 9; Accepted 2022 Oct 28
Copyright: ©2022 Greif et al.
Copyright year: 2022
Copyright holder: Greif et al.
License: This is an open access article distributed under the terms of the Creative Commons Attribution License, which permits unrestricted use, distribution, reproduction and adaptation in any medium and for any purpose provided that it is properly attributed. For attribution, the original author(s), title, publication source (PeerJ) and either DOI or URL of the article must be cited.
License URL: https://creativecommons.org/licenses/by/4.0/

Keywords: Early vertebrates, Meckel’s cartilage, Anti-atlas, Morphometrics, Shape analysis, Morphospace, Hangenberg black shale, Devonian

Funding: Swiss National Science Foundation 200020_184894 Generalitat Valenciana APOSTD/2021/119 Merle Greif and Christian Klug received financial support from the Swiss National Science Foundation (project nr. 200020_184894). Humberto Ferrón is funded by the Generalitat Valenciana (APOSTD/2021/119). The funders had no role in study design, data collection and analysis, decision to publish, or preparation of the manuscript.

==============================
Fossil chondrichthyan remains are mostly known from their teeth, scales or fin spines only, whereas their cartilaginous endoskeletons require exceptional preservational conditions to become fossilized. While most cartilaginous remains of Famennian (Late Devonian) chondrichthyans were found in older layers of the eastern Anti-Atlas, such fossils were unknown from the Hangenberg black shale (HBS) and only a few chondrichthyan teeth had been found therein previously. Here, we describe a Meckel’s cartilage from the Hangenberg black shale in Morocco, which is the first fossil cartilage from these strata. Since no teeth or other skeletal elements have been found in articulation, we used elliptical Fourier (EFA), principal component (PCA), and hierarchical cluster (HCA) analyses to morphologically compare it with 41 chondrichthyan taxa of different size and age and to evaluate its possible systematic affiliation. PCA and HCA position the new specimen closest to some acanthodian and elasmobranch jaws. Accordingly, a holocephalan origin was excluded. The jaw shape as well as the presence of a polygonal pattern, typical for tessellated calcified cartilage, suggest a ctenacanth origin and we assigned the new HBS Meckel’s cartilage to the order Ctenacanthiformes with reservations.

Introduction

Fossil chondrichthyans (sharks, rays and chimaeroids) are mainly known from the Devonian onward (Brazeau & Friedman, 2015). Exceptional, putative chondrichthyan, as well as acanthodian finds date back to the Silurian (Burrow & Rudkin, 2014; Andreev et al., 2016). Only teeth, scales and fin spines of chondrichthyans (whole group, including acanthodians) are strongly mineralized while chondrichthyan endoskeletons are predominantly made of unmineralized cartilage that is only rarely preserved (Seidel et al., 2020).

Despite the difficulties of preservation, chondrichthyan skeletons are frequently found in the middle and late Famennian strata in the Tafilalt and Maïder regions of southern Morocco and constitute important contributions to the understanding of early vertebrates (Ginter, Hairapetian & Klug, 2002; Derycke et al., 2008; Frey et al., 2018; Frey et al., 2020). However, in the late Famennian Hangenberg black shale layers of Morocco, nearly no vertebrate remains have been collected or described so far. The only known contributions to the vertebrate fossil record that are known from these strata are a few chondrichthyan teeth, which are not described but only mentioned in the literature (Klug et al., 2016; Frey et al., 2018) as well as some chondrichthyan ichnofossils from layers just above the Hangenberg black shale (basal Hangenberg Sandstone; Klug et al., 2021). Here, we describe a lower jaw found in the Anti-Atlas that represents the first reported cartilaginous remain from the Moroccan Hangenberg black shale.

Outcrops of sediments that were laid down in the time around the end-Devonian Hangenberg crisis can be found at many localities of the Tafilalt and Maïder regions of the Anti-Atlas (Kaiser et al., 2011; Kaiser, Aretz & Becker, 2015; Klug et al., 2021). The Hangenberg crisis was a global mass extinction event at the Devonian/Carboniferous boundary (Caplan & Bustin, 1999; Kaiser et al., 2011), which reflects one of the six largest mass extinction events in earth’s history. The Hangenberg crisis followed the Kellwasser event at the Frasnian/ Famennian boundary and affected vertebrate groups to an extent that is comparable to the Big Five mass extinctions (McGhee, 1996; McGhee et al., 2012; McGhee et al., 2013). Therefore, it is seen as a bottleneck in vertebrate evolution and the recovery of formerly diverse vertebrate groups (such as some agnathans, sarcopterygians and placoderms) after the event was minimal (Sallan & Coates, 2010; Frey et al., 2018). Indeed, the Hangenberg crisis was more severe than formerly thought and caused a larger diversity loss on genus level than the Kellwasser event (Sallan & Coates, 2010). The Hangenberg black shale marks the main extinction phase of the event and was laid down during a supposed global transgression linked with widespread anoxia. The anoxia were likely caused by eutrophication that led to global extinctions in numerous invertebrate groups such as algae, sponges, ammonoids, trilobites, brachiopods and bivalves (Algeo & Scheckler, 1998; Sallan & Coates, 2010; Kaiser et al., 2011; Kaiser, Aretz & Becker, 2015) but also in vertebrate groups like chondrichthyans and placoderms (Kaiser et al., 2011).

While remains of the previously mentioned invertebrate groups are quite common in the Hangenberg black shale (Clausen et al., 1924; Marynowski et al., 2012; Klug et al., 2016; Zhang et al., 2019), it lacks vertebrate remains, which makes the new Meckel’s cartilage a particularly important fossil.

The cartilaginous endoskeletons of chondrichthyans are covered by a thin layer of calcified cartilage (Kemp & Westrin, 1979; Dean & Summers, 2006; Seidel et al., 2016; Seidel et al., 2020; Maisey et al., 2020). This thin layer typically shows a distinct polygonal pattern, which is caused by the presence of tesserae, namely the tessellated calcified cartilage (Seidel et al., 2016; Seidel et al., 2020; Seidel, Jayasankar & Dean, 2021; Maisey et al., 2020). Such cartilage is characteristic for extant as well as Devonian crown chondrichthyans (elasmobranchs and holocephalans; Long et al., 2015; Maisey et al., 2020) while these polygonal structures tend to be less distinct in acanthodians (a paraphyletic group of stem chondrichthyans; Rücklin et al., 2021), where only subtessellated calcified cartilage or globular calcified cartilage is reported (Dean & Summers, 2006; Brazeau & Friedman, 2014; Brazeau et al., 2020; Maisey et al., 2020). Globular calcified cartilage builds the inner layer of tessellated calcified cartilage and can build the entire hard tissue. If globular calcified cartilage is present on the surface a granular pattern is to be expected (Burrow et al., 2015; Maisey et al., 2020). Subtessellated calcified cartilage shows fissures along the surface, which result in an unorganized pattern. Tessellated calcified cartilage with an outer prismatic layer, in contrast, is well organized and a polygonal pattern is distinct (Maisey et al., 2020; Seidel et al., 2020).

Among the cartilaginous remains, jaws are one of the most relevant anatomical structures from an evolutionary perspective. The evolution of jaws, the Meckel’s cartilage, is seen as a key innovation of gnathostomes enabling the first gnathostomes to broaden their range of feeding strategies and prey upon a much greater diversity of animals (DeLaurier, 2019; Deakin et al., 2022). These innovations contributed greatly to the radiation of gnathostomes and possibly to the decline of agnathans (Brazeau & Friedman, 2015; Hill et al., 2018). Nevertheless, only very few quantitative studies about jaw shapes have been published. For example, Hill et al. (2018) quantified jaw shape in extant and in Palaeozoic fishes (Chondrichthyes, Sarcopterygii, Actinopterygii, Placodermii, Acanthodii) and demonstrated that jaw shape has a greater disparity in extant fish clades than during the early gnathostome radiation (Silurian and Devonian). This is mostly caused by the great morphological disparity among extant actinopterygians (Hill et al., 2018). Deakin et al. (2022) mentioned an increasing disparity in jaw shape with ongoing evolution but the functional disparity of early vertebrate jaws to be highest very early in jaw evolution and optimized for a predatory function. Anderson et al. (2011) also deals with jaw disparity and the influence of environmental changes such as the Kellwasser event, which does not seem to affect jaw disparity very much.

The phylogenetic relations within the chondrichthyan total group are still a widely discussed topic (Hanke & Wilson, 2006; Brazeau, 2009; Davis, Finarelli & Coates, 2012; Burrow & Rudkin, 2014; Brazeau & Friedman, 2015; Brazeau & De Winter 2015; Giles, Friedman & Brazeau, 2015; Qiao et al., 2016) and acanthodians were just recently recognized as a paraphyletic goup of stem chondrichthyans (Zhu et al., 2013; Coates et al., 2017; Rücklin et al., 2021). Members of this group show characteristics of both principal clades of living gnathostomes (chondrichthyans and osteichthyans), are covered with scales and are often referred to as “spiny sharks” because of the spines in front of their dorsal, anal and paired fins as evident in most taxa of this group (Miles, 1970; Miles, 1973; Burrow & Rudkin, 2014; Qiao et al., 2016). The relationship between jaw shape and phylogeny remains an elusive question since ecological factors likely influence jaw shape to a great degree as well.

Our main aim in this article is (1) to give a detailed description of this novel find and (2) to determine its possible systematic affiliation. For the latter, we used geometric morphometrics since the Meckel’s cartilage was found isolated with no further skeletal parts, teeth or scales associated and is therefore hard to assign to a specific taxon. We applied elliptical Fourier (EFA), principal component (PCA) and hierarchical cluster analyses (HCA) to the new small Meckel’s cartilage and 41 more chondrichthyan and acanthodian lower jaws. By this action, a morphospace is created which is informative about the relationship between lower jaw shape and phylogeny.

Materials & Methods

The specimen PIMUZ A/I 5139 (Fig. 1) was found in the Moroccan Anti-Atlas at the locality Madene El Mrakib (N30.73093°, W4.70749°). Permit for fossil collection and export were given by the Ministère de l’Energie, des Mines, de l’Eau et de l’Environnement, Rabat, Morocco. The specimen is stored at the Palaeontological Institute and Museum of Zurich (Switzerland). It was largely exposed, but covered parts were carefully prepared using a thin steel-needle. Photos of the specimen showing its shape, proportions and preservation (Fig. 1) were taken using a Nikon D2X. Colour and contrast were slightly adjusted in Adobe Photoshop (Adobe, Inc., San Jose, CA, USA). To show the structure of the fossil’s surface in more detail, close-ups were taken with a Leica MZ16 F microscope (Figs. 1C, 1D and 1E) and gently adjusted in colour and contrast as well.

Figure 1 Meckel’s cartilage outlines and close ups.

Meckel’s cartilage of a ctenacanth chondrichthyan from the Hangenberg black shale, Madene El Mrakib; PIMUZ A/I 5139. A, lateral view; B1, traced outline and ventral ridge; B2, counterpart with outline; C, Close up of the anterior area; D, close up of the posterior area; E1,2, Close-up photos of the cartilage showing the polygonal pattern. Abbreviations: sym, symphysis; ma, muscle attachment area; vr, ventral ridge; re.fl, retroarticular flange. Scale bar for A, B1,2 equals five mm. Scale bar for C, D, E1,2 equals one mm. Arrow indicates Anterior (A) and Posterior (P).

Morphometrics

Morphometric techniques together with multivariate and cluster analysis are standard methods to quantify morphology and evaluate groupings or affinities among taxa (Kaesler & Waters, 1972; Younker & Ehrlich, 1977; Ferrario et al., 1999; Daegling & Jungers, 2000). Here, we use morphometric analyses to compare the new isolated Meckel’s cartilage to shapes of other lower jaws with known systematic affiliation and find the most similar shape, or group of shapes, to help determine the new Meckel’s cartilage taxonomic affinities at least approximately. To carry out the analyses, outlines of 41 lower jaws representing the main stem and crown chondrichthyan orders were drawn based on photographs and illustrations from the literature (Appendix 1) using the vector-based software Affinity Designer (Serif, Nottingham, Nottingham, United Kingdom). Sampling is constrained by the limited number of well-preserved fossils of Meckel’s cartilages. The jaw shapes used in the analysis were chosen based on the quality of preservation and completeness of the Meckel’s cartilage as could be seen in the publications. The sampled jaws belong to taxa from different periods and localities and cover a wide range of sizes (Appendix 2). This broad sampling range (regarding time, locality and size) was used to find general differences in shape between the different groups.

All Meckel’s cartilage outlines were digitized in TPS software (Rohlf, 2015). Elliptic Fourier Analysis (EFA) was then performed in the Momocs package (Bonhomme et al., 2014) in R (R Development Core Team, 2020) to statistically compare all sampled lower jaw shapes. A total number of 25 harmonics were considered, which gather nearly 99% of the cumulative harmonic power (seen as a measure of shape information) and reconstructs actual morphologies with high accuracy. We obtained a virtual morphospace by performing a principal component analysis (PCA, Fig. 2) on the preordination data to plot the main shape variations. To quantify the morphological similarity amongst the studied jaws, a Hierarchical Cluster Analysis (HCA) using the R package ‘dendextend’ (Galili et al., 2019) was conducted. Phylogenetic signal was assessed using the lambda and K statistic with 1,000 random permutation in the R package ‘phytools’ (Revell, 2012). Additionally, a Mantel test, correlating phenetic (morphological) and phylogenetic distances was performed in order to assess the degree of morphological convergence in our sample. These metrics are expected to show greater decoupling and, consequently, lower correlation where homoplasy occurs. Phenetic distances were calculated as Euclidean distances in the morphospace, considering all PCs. We repeated the tests in a set of 1,000 phylogenetic trees that accounted for phylogenetic and stratigraphic uncertainty. The tree topology is based on ongoing research by Klug and colleagues (C Klug, M Coates, I Frey, M Greif, M Jobbins, A Pohle, A Lagnaoui, W Bel Haouz, unpublished data, 2022). Polytomies were randomly resolved 1,000 times and each resulting tree was calibrated by randomizing the tip age of every species within the chronostratigraphic unit, at age or subperiod rank, where their first appearance occurs, using the R package ‘paleotree’.

Figure 2 PCA and morphospace showing all sampled lower jaws.

Principal Component Analysis of some fossil and recent chondrichthyan lower jaws. Orange colours: acanthodians; purple colours: holocephalan; blue colours: elasmobranchs. The new lower jaw from the Hangenberg black shale is represented by a black dot and grey colours represent lower jaws of unknown class and order. A jaw morphospace is represented in the background showing the shape variation. The new Hangenberg black shale jaw plots close to jaws of acanthodians as well as elasmobranchs. Lv, Latviacanthus ventspilsensis; Is, Ischnacanthus sp.; Po, Palidiplospinax occultidens; Dh, Dracopristis hoffmanorum; Ct, Ctenacanthus sp.; Hd, Heslerodus divergens.

Results

Systematic palaeontology

Class Chondrichthyes Huxley, 1880	
Subclass ? Elasmobranchii Bonaparte, 1838	
Order ? Ctenacanthiformes Glikman, 1964	

The Meckel’s cartilage with a total length of 18 mm and a height of up to six mm is nearly complete and preserved in lateral view (Fig. 1A). The posterior part is somewhat incomplete in the main plate and entirely missing in the counterpart (Figs. 1A, 1B and 1D). While most of the specimen is visibly different from the sediment due to its internal structure and colour, in the posterior part the Meckel’s cartilage limits are less clear and the exact borders between fossil and sediment are difficult to determine. The specimen shows a bright grey to white colour and most of it is somewhat brighter than the sediment. In the main fossil plate and in the counterpart, a distinctive polygonal pattern of the calcified cartilage is visible mainly in the posterior part (Fig. 1E1,2) while in the middle to anterior part, the specimen is mineralized in a bright colour. The tessellation is not as geometric as in some extant species (Seidel et al., 2020; Seidel, Jayasankar & Dean, 2021) but the polygons are distinct. In some areas, the borders of the polygonal tesserae are clearly distinguishable by white outlines that most likely represent the intertesseral fibres (Seidel et al., 2016). Even though the tesserae borders are distinct, the corners, as well as the borders in general are rounded and less distinct. Despite the blurriness, the pattern is very similar to the one that can be seen in the crown chondrichthyan Tristychius arcuatus (Brazeau & Friedman, 2014, Fig. 5C and D).

The ventral edge of the Meckel’s cartilage is gently convexly curved. The ventral ridge is discernible in spite of the compaction, especially in the middle to posterior part. It follows the shape of the outline of the jaw until about 2.5 mm distance from the posterior end when it bends upwards (Fig. 1A). The Meckel’s cartilage becomes higher from posteriorly until just before the articulation. It displays one bulge at the thickened anterior end, which is about four mm long and might represent the symphysis. This bulge is followed by a shallow depression, which is 3.5 mm long and a shallow bulge of about 2.5 mm length. The preservation is insufficient to identify muscle attachments with confidence. We assume that the anterior nine mm was the tooth-bearing part (dental sulcus) because the concave upper edge anterior to the articulation ends there and it appears like the dorsal side broadens from this point anteriorly. The next depression extends over 7.5 mm and ends at the articulation. Although the specimen is flattened, the retroarticular flange (cf. Long et al., 2015) at the posterior end is still preserved as a knob. The articulation is positioned dorsally in the posterior end of the jaw but unfortunately, the preservation does not allow to determine the exact shape of the articulation and it seems incomplete.

Morphometric analyses

The PCA shows clear separation between the jaws of the two chondrichthyan clades Elasmobranchii and Holocephalii (Fig. 2). PC 1 (59% of variance) is mostly related to changes in jaw thickness with decreasing thickness from negative to positive scores. PC 2 (13% of variance) mainly reflects changes of the jaw curvature (from strongly convex to slightly concave), with a decrease in curvature from negative to positive scores (Fig. 2). PC 3 (6% of variance) mostly describes changes in the curvature of the anterior end of the jaw as well as changes of the roundness of the posteroventral edge of the jaw (Fig. 2). Holocephalan jaws occupy high PC1 scores of about 0.05 to 0.17 and positive PC2 scores and show relatively slender and only slightly curved morphologies. Elasmobranch jaws occupy a wider score range with PC1 scores between −0.8 to 0.08 and PC2 scores between 0.07 and 0.10 (Fig. 2). Most of them plot in the centre of the morphospace between PC1 scores of about −0.5 and 0.01 and PC2 scores around 0.0. Elasmobranch jaws show greater shape variation than holocephalan jaws, from thick and bulky to relatively slender. Acanthodian jaws occupy PC1 scores from −0.11 to 0.10 and PC2 scores of −0.12 to 0.05 (Fig. 2) and overlap to a large extent with elasmobranch and holocephalan jaws. Acanthodian jaw shapes vary from bulky and curved to slender and straight. The new specimen plots at −0.01/0.025 (PC1/PC2), which is close to the other sampled acanthodians and some elasmobranchs. The new specimen plots closest to the acanthodian taxa Ischnacanthus sp. and Latviacanthus ventspilsensis. Furthermore, some ctenacanths plot very close: Dracopristis hoffmanorum, Ctenacanthus sp. Heslerodus divergens, as well as another elasmobranch of the order Synechodontiformes: Palidiplospinax occultidens (Fig. 2).

In the dendrogram derived from the HCA, the new Hangenberg black shale Meckel’s cartilage plots closest to the acanthodian Latviacanthus ventspilsensis. The acanthodian Ischnacanthus sp. and the elasmobranch of Heslerodus divergens constitute sequential sister groups to those two (Fig. 3). Overall, there is not a clear grouping among the three classes (Fig. 3). However, at a lower clustering rank, a separation between holocephalans and elasmobranchs is supported while acanthodians plot together with either elasmobranchs or holocephalans (Fig. 3). We find a significant phylogenetic signal as measured by the metrics K (equal to 0.501 ± 0.071; p-value = 0.004 ± 0.004) and lambda (equal to 0.995 ± 0.123; p-value = 0.0001 ± 0.0001; Fig. 4), but no significant correlation in between phenetic and phylogenetic distances in the Mantel tests (R statistic = −0.045 ± 0.009; p-value = 0.632 ± 0.032, all data expressed in mean ± standard deviation, Fig. 5).

Figure 3 Dendrogram showing morphological distances of the sampled lower jaws.

Dendrogram showing morphological distances regarding the first principal components from the PCA. Orange colours: acanthodians; purple colours: holocephalan; blue colours: elasmobranchs. The elasmobranchs plot mainly on the top, while holocephalan jaws plot mainly at the bottom. Acanthodian jaws are scattered over the whole dendrogram. The lower jaw from the Hangenberg black shale is closest to some acanthodian jaws such as that of Ischnacanthus sp.

Figure 4 Phylogenetic signal metrics and tests of significance.

Phylogenetic signal metrics and tests of significance performed in 1,000 trees accounting for phylogenetic and stratigraphic uncertainty.

Figure 5 Mantel test results.

Mantel test analysis performed in 1,000 trees accounting for phylogenetic and stratigraphic uncertainty. R statistic values close to 1 or −1 support strong correlation between phylogenetic and phenetic distances, while values close to 0 support weak correlation.

Discussion

Our methodological framework based on EFA, PCA and HCA allows for discriminating holocephalans from elasmobranchs as well as some clades of lower systematic rank, but discrimination of acanthodians as a whole from holocephalans and elasmobranchs is not evident (Figs. 2 and 3). We detect a strong phylogenetic signal in our dataset (Fig. 4), altogether suggesting that outline jaw shape by itself can be, to some extent, informative for systematic placement of disarticulated remains and add support to other evidence. However, it has to be kept in mind that our morphometric analysis considers two-dimensional outline shape and, potentially, some relevant anatomical information to discriminate among other groups might not be captured. Further, the lack of correlation between phylogenetic and phenetic distances in Mantel tests (Fig. 5) entail the presence of an important homoplasy, which might hinder the interpretations of phylogenetic affinity from general morphology. Similarities in jaw shape can also result from adaptation. Jaw shape can, for example, be an adaption to a certain lifestyle as in durophagous sharks (Herbert & Motta, 2018) or in general be connected to diet in combination with water depth (Motta & Huber, 2012). Small variations in shape could also occur due to fossilisation, preparation and errors in redrawing the different outlines, but we do not expect this to have a major effect in our results as preliminary studies have supported that biological signal is still well preserved when minor taphonomical alterations exist (Angielczyk & Sheets, 2007).

The inclusion of the new Hangenberg black shale jaw in the analysis revealed that it is most similar in shape to lower jaws of certain acanthodian (i.e., Ischnacanthus sp. and Latviacanthus ventspilsensis) as well as elasmobranchs (the ctenacanths Dracopristis hoffmanorum, Ctenacanthus sp., and Heslerodus divergens; and the synechodontiform Palidiplospinax occultidens) (Figs. 2 and 3). A holocephalan affinity is unlikely as all considered taxa from this group fall in a separate area of the morphospace. The Hangenberg black shale jaw sits slightly closer to acanthodian jaw shapes than to elasmobranch jaw shapes but whether it is of acanthodian or of elasmobranch origin is difficult to ascertain solely from those analyses and further information is needed to determine its possible origin. An acanthodian origin would entail its inclusion within the paraphyletic groups of stem chondrichthyans (Rücklin et al., 2021) while an elasmobranch origin would entail its inclusion in one of the two sister groups of crown chondrichthyans (Elasmobranchii and Holocephali, (Maisey, 2012).

Besides the HBS Meckel’s cartilage, the only vertebrate fossils known from the Hangenberg black shale are some poorly preserved chondrichthyan teeth (Klug et al., 2016), which are not determined but could be of symmoriiform origin (? Stethacanthus, Coates & Sequeira, 2001, Figs. 5F and 5I). However, given out analyses, a holocephalan origin seems unlikely. The exclusion of a holocephalan origin is further supported by the absence of a terminally positioned articulation, which is typical for holocephalans (Coates et al., 2017, character matrix). Due to incomplete preservation of the articulation, it cannot be compared in detail to other chondrichthyan lower jaws.

Among the few characters present in the new HBS Meckel’s cartilage, some can help to further distinguish its most probable affinity. Thus, the jaw of the ctenacanth Heslerodus divergens (Hodnett et al., 2021) seems to share some features not directly captured by outline analysis, which are less distinct in both acanthodian jaws that plot close to the HBS jaw. The jaw of Heslerodus divergens has a relatively thin anterior to middle part comparable to the first nine mm of the new jaw that we described as the probable tooth bearing part. Following this, in both jaw shapes, a ridge is present leading to a second depression that ends in the articulation. In the jaw of Heslerodus divergens, this shape is more distinct than in the HBS jaw while both acanthodian jaws are dorsally straighter shaped (Fig. 6). Additionally, Hodnett et al. (2021) describes “a well-developed ventral ridge on the lateral margin of the Meckel’s cartilage that extends over two thirds the length of the jaw(…)”, as a synapomorphy of ctenacanths. A ventral ridge is one of the few features of the new Hangenberg black shale jaw, which is easily recognized (Fig. 1). Ischnacnathus sp. shows a ventral ridge as well but when comparing the HBS jaw ventral ridge to the other two, the one of Heslerodus divergens is a lot more similar (Fig. 6).

Figure 6 Visual jaw shape comparison.

Direct comparison of the new HBS Meckel’s cartilage (grey, top) with the two most similar jaw shapes of two different groups (pink, middle) and an overlay of both (pink and grey, bottom). A, the elasmobranch Heslerodus divergens. B, the acanthodian Ischnacanthus sp. Different characteristic points, that were not captured by the PCA directly, as well as the ventral ridge are compared and both shapes are shown in overlap with the HBS Meckel’s cartilage.

In addition, a distinct polygonal structure is visible on the surface of the jaw (Fig. 1C). This pattern is characteristic for tessellated calcified cartilage, which is widely accepted as a synapomorphy of extant and extinct crown chondrichthyans (Brazeau & Friedman, 2014; Long et al., 2015; Seidel et al., 2016; Seidel, Jayasankar & Dean, 2021; Maisey et al., 2020). Tessellated calcified cartilage is made of an inner layer of globular calcified cartilage and an outer layer of prismatic calcified cartilage (Maisey et al., 2020). Only the outer prismatic layer shows the typical polygonal pattern while the globular calcified cartilage shows a granular surface (see for example the acanthodian Climatius reticulatus in Burrow et al., 2015, Fig. 1I).

Fossils of the acanthodian group (paraphyletic group of stem chondrichthyans) mostly do not show a polygonal pattern, since no prismatic outer layer is present, but only globular calcified cartilage (Maisey et al., 2020). However, Maisey et al. (2020) describes the presence of subtessellated calcified cartilage in some acanthodians, while actual tessellated calcified cartilage (showing the outer prismatic layer) is apparently absent (Brazeau & Friedman, 2014). Acanthodians like Climatius (Burrow et al., 2015), Ischnacanthus (Burrow et al., 2018) or Cheiracanthus (Den Blaauwen, Newman & Burrow, 2019) are mentioned to show this subtessellated calcified cartilage. When looking at Climatius, it appears granular and no actual polygons are visible on the surface as mentioned above (Burrow et al., 2015, Fig. 1I). In Ischnacanthus (Burrow et al., 2018), a subtessellated calcified cartilage is described using histology; we cannot compare the HBS specimen to that. In Cheiracanthus (Den Blaauwen, Newman & Burrow, 2019), the surface appears “globular or randomly tessellated”. To sum this up, acanthodian fossils, or stem chondrichthyans, show a rather globular or irregular pattern (Burrow et al., 2015, Fig. 1I; Long et al., 2015, Fig. 9A), which differs a lot from the regular polygonal pattern in crown chondrichthyans.

A polygonal pattern is evident in the new specimen but the borders of the single tesserae are slightly blurred taphonomically, which might have been caused by dissolution of the unmineralized collagen between the tiles (intertesseral fibre; Seidel et al., 2016). However, the pattern is distinct and regular, making an elasmobranch origin more likely than an acanthodian origin. In fact, it is as regular as the polygonal pattern in the crown chondrichthyan Tristychius arcuatus (Brazeau & Friedman, 2014).

Based on the results from morphometric analyses and the presence of both a ventral ridge on the lateral margin and tessellated calcified cartilage with a regular polygonal pattern, we assign the new Meckel’s cartilage to the order Ctenacanthiformes with some reservations (Fig. 7). To some degree, this classification remains tentative and a bigger sample size could help to test the hypothesis. Further fossil finds as well as a better understanding of the early development of tessellated calcified cartilage in early fishes could help to classify the new jaw in more detail. However, this study presents an important fossil find, filling a gap in the fossil record and provides crucial information about the difficulties of determining the systematic affiliation of isolated cartilaginous fossil remains.

Figure 7 All sampled outlines in a phylogenetic tree showing the possible position of the new Meckel’s cartilage.

Simplified chondrichthyan phylogeny modified after Klug et al. (in prep.). The lower jaw from the Hangenberg black shale is figured together with the taxa used in the Fourier Analysis. The shapes of the lower jaws were redrawn from the literature (Appendix 1). The new HBS jaw is suggested to be of ctenacanthiform origin regarding the analyses and comparison of characters.

Conclusions

The newly described Meckel’s cartilage is the first known fossil cartilage remain from the Hangenberg black shale from the Moroccan Anti-Atlas. It is 18 mm in length, ventrally convexly curved and shows a biconcave dorsal edge. PCA and HCA reveal a strong similarity in shape with certain acanthodians and elasmobranchs and a phylogenetic signal is detected in our dataset. We conclude that jaw shape can be informative about the systematic placement of disarticulated skeletal elements but further information is needed since homoplasy is pervasive. The structure of the tessellated calcified cartilage was used as a character for classification. It shows a distinct polygonal pattern which is characteristic for crown chondrichthyans.

Furthermore, its general shape as well as the shape of the ventral ridge were compared to two of the jaws that were classified as the most similar by PCA and HCA analyses. This comparison suggests a ctenacanth affiliation. Considering the mentioned evidence, we assigned the new lower jaw to the order Ctenacanthiformes, tentatively.

Supplemental Information

Supplemental Information 1 List of all sampled lower jaws and the corresponding literature

Click here for additional data file.

Supplemental Information 2 R code

Click here for additional data file.

At an earlier stage, Louis Dudit (Zürich) helped with the Fourier analysis. We showed photos of the Meckel’s cartilage to Carole Burrow (Queensland) and Jake Leyhr (Uppsala) and discussed its affiliation. We greatly appreciate their suggestions regarding both the jaw and the teeth from the HBS. We thank the reviewers for carefully reviewing our manuscript and for, thereby, helping to improve it.

Additional Information and Declarations

Competing Interests

Author Contributions

Field Study Permissions

Data Availability

The authors declare there are no competing interests.

Merle Greif conceived and designed the experiments, performed the experiments, analyzed the data, prepared figures and/or tables, authored or reviewed drafts of the article, and approved the final draft.

Humberto G. Ferrón conceived and designed the experiments, performed the experiments, authored or reviewed drafts of the article, and approved the final draft.

Christian Klug conceived and designed the experiments, analyzed the data, authored or reviewed drafts of the article, and approved the final draft.

The following information was supplied relating to field study approvals (i.e., approving body and any reference numbers):

The Ministère de l’Energie, des Mines, de l’Eau et de l’Environnement (Direction du Développement Minier, Division du Patrimoine, Rabat, Morocco) approved the study.

The following information was supplied regarding data availability:

All R code files that were used to create Fourier Analysis/ PCA plots and further analyses are available in the Supplemental File.

The specimen (PIMUZ A/I 5139) is stored at the Palaeontological Institute and Museum of Zurich (Switzerland).

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
