# Peer review of "A new Meckel’s cartilage from the Devonian Hangenberg black shale in Morocco and its position in chondrichthyan jaw morphospace"

_PeerJ, doi:10.7717/peerj.14418_

## Round 0.1 · original submission · Major Revisions

Dear authors,
I am glad to inform you that we have now three review reports for your article entitled “A minute Meckel’s cartilage from the Devonian Hangenberg black shale in Morocco and its position in chondrichthyan jaw morphospace”, and all are favorable to recommend publication after some suggested changes be incorporate in order to improving the relevance of this study.

I would be grateful if you consider all the requests from the reviewers because I think they are very interesting approaches that surely will strengthen your conclusions.

For instance, it is fundamental that you define the main goal/s of this contribution. It may be the description of a new and novel fossil from the Hangenberg black shale, which is problematic to assign to a taxonomic category. Following this line of reasoning you have to be consistent with the results obtained from the selected applied methodology, which as reviewer 1 pointed out was useful to resolve the morphologic study but not so to determine the phylogenetic relationships of the described specimen. That said, I have to note that I felt a little lost trying to follow your discussion about the affinities of the fossil.

Another possibility can be that the description of the Meckel’s cartilage found could contribute to enhancing the relevance of the Hangenberg event on the chondrichthyan evolution, as was suggested by Reviewer 2.

Finally, another additional aim of your manuscript which is also an important goal can be to show the problem around the study of a poorly preserved, unique and incomplete fossil from which insufficient data were provided by the analyses performed to solve confidently its affinities right now, but leaving a high expectation for future research, as commented by Reviewer 3.

I really think that all of these aspects can be the aim of this article, but you have to develop them in depth for the following new version of this manuscript.
Other suggestions and comments from the reviewers will need your careful consideration and attention because they are key changes that will highly improve this work, particularly the changes requested for the figures.

Hoping you find useful these clear, complete and constructive contributions from the reviewers, I look forward to seeing the revised new version of this interesting manuscript soon.
Very best wishes,
Graciela Piñeiro

Reviewer 1 ·

Basic reporting

The paper entitled “A minute Meckel’s cartilage from the Devonian Hangenberg black shale in Morrocco and its position in chondrichthyan jaw morphospace” is well written and presents an important new fossil occurrence for the Late Devonian strata of Morrocco. The preservation of the analyzed specimen, as stated by the authors, does not allow a clear taxonomic identification and thus its affinities were attempted to be established on morphometric grounds. The literature cited is up-to-date and is sufficient for the background provided and for the discussion. Figures are inteligible and generally show enough information to support the claims made in the manuscript (but see comments below).
Although this paper presents important data and the methodology used has potential to help understanding the diversity of jaw morphologies in chondrichthyans—but maybe not phylogenetic affinities—there are still several adjustments that need to be done before this paper can be accepted for publication.

Experimental design

Although I understand why the authors chose the approach for this manuscript, there are aspects of the methods that need to be better explained. First, considering that the morphometric analysis used here was meant to help identify a potential phylogenetic placement for the analyzed specimen, I fail to understand how the authors expect this approach to work (besides separating Holocephalans and Chondrichthyans) just by reading the methods sections. I would recommend adding more detail to this section on why such methods were used and if similar approaches have been tested before with other groups and yielding significant results supported by other data on phylogenetic placement of taxa.
Additionally, I would like to see a brief explanation on how the taxon sampling was done. Were the jaws selected to cover the most possible of the chondrichthyan phylogeny? Or were they sampled based on preservation/availability? A mix of these two? If sampling was based on the tree of Klug et al. (in prep) this should be made explicitly.
There are taxa of similar morphology and age that are not included in the analysis that have well-preserved Meckel’s cartilages described in the literature. A good example is the early chondrichthyan Antarctilamna ultima from the Famennian black shales of the Witpoort Formation in South Africa (Gess, R. Coates, MI. 2014. High-latitude chondrichthyans from the Late Devonian (Famennian) Witpoort Formation of South Africa. Palaontol Z.)

Validity of the findings

The findings presented in this manuscript are valuable contributions to the field, especially the occurrence of the fossil itself. The methods used also have potential to be a valuable methodology for better understanding isolated remains with few characters that are informative for taxonomy. However, as stated above in the experimental design section, there needs to be better detailing on how the sampling was done, and why this methods were chosen in the first place. All data is consistent and the product of good scientific work and was made available for review. The code works and provides the same results presented in the manuscript (but see minor comment on the “additional comments” section below). Conclusions are well-written and are supported by the results and discussion presented by the authors.

Additional comments

INTRODUCTION:
The introduction is well-written and provided useful background on the record of chondrichthyan jaws in the Paleozoic and biases of preservation that favor teeth, dermal denticles and spines. The section on the position of acanthodians is also sound and helps situate the reader. My only suggestion would be including a paragraph on the diversity of chondrichthyan articulated elements from Paleozoic strata and how these are more common than similar remains in post-Paleozoic strata.

Line 50-51: Explicit mention to these lineages should be added here, to make text clearer for non-specialists. I assume you are referencing the two extant gnathostome lineages, Chondrichthyes and Osteichthyes.

Line 52: climatiids also have spines that are not in direct association with fins. Also, maybe you could cite some of the classic literature on acanthodians here such as Miles, 1965.

RESULTS:
The description needs to be written in more detail. For example, you mention a "distinctive polygonal pattern" for the calcified cartilage but do not describe this pattern.
The jaw articulation is partially visible on the posterior end. Is it possible to describe it?
Any indication of the geometry of the glenoid fossa?
Which muscle attachment are you mentioning when talking about the symphyseal region?
Is the jaw preserved showing the mesial or the lateral surface?
You mention that no tooth is found in association to this jaw. It is hard to tell only from the images provided. I assume the markings along the dorsal margin of the jaw are preparation marks? But disarticulated from the jaw in Fig.1A there what seems like a bicuspid tooth? Or is that also an artifact of the sample or preparation?
Is there any evidence that the margins of this specimen are the real margins of the outline of the jaw? Could any part be obscured by the matrix?

Line 198: which muscle attachment are you referring to here?

DISCUSSION:
The discussion presented by the authors can be divided into three parts. On the first part they discuss the experimental results presented above and the limitations of the methods used/available. The second part focuses on comparing the morphology of the new jaw to the taxa that plotted close to it in the PCA and HCA. The third and last part comments on potential indicatives of the crown chondrichthyan affinities of this specimen based on the superficial morphology of the calcified cartilage. Although these sections are well-written and organized in a coherent manner, there is room for including a new paragraph on previous work that tries to determine the affinities of fossil chondrichthyans based on disarticulated elements, especially jaws, stating potential characters that have been used or inferring new characters that might elucidate the affinities of poorly know taxa based on the results presented above. Does the morphology of the lower jaw outline provide sufficient information for taxonomic assessment? What features of a Meckel’s cartilage are important for identification when found in isolation? Some of these issues are briefly discussed when the authors compare the specimen to other taxa, but I believe a separate paragraph better detailing this would be a good addition to the manuscript (consider this only a suggestion).

Line 279-280: Here you state that there is a ventral ridge in your specimen that indicates an affinity to ctenacanthiformes, but this feature is not described in detail in the results section (it is only stated that it’s present). If this feature is useful for taxonomic identification I believe it should get a better description than what is provided here.

FIGURES:
Fig. 1 – I would remove the scale bar length numbers from the figure and include these in the caption.
Fig. 2 – The distinction between holocephalans and elasmobranchs described in the results is indeed present here but is still some considerable overlap between the two. Could be interest to provide in the discussion if there might be ways to better distinguish between these two groups with morphometric data. Would using 3D data help?
Fig. 3 – This dendrogram seems to indicate that the morphology of these jaws based on the outline is not indicative of phylogenetic affinity. Even the distinction between holocephalans and elasmobranchs described above and shown in previous figures is not evident here.

CODE:
Not sure if I was doing something wrong, but when checking the code I could not get function ‘grid.arrange’ to work without calling the package gridExtra, which is not listed at the beginning of the script.

·

Basic reporting

The English is good, although there are a few examples of clunky sentence structure that could be fixed with a careful reread. References are good, as is the structure of the article and the article has relevant results.

Context is generally good, but I'd recommend providing a bit more context re. Hangenberg sharks. In its current form the introduction lacks focus: the introduction kicks off very generally with a discussion of early vertebrate origins and how sharks make bad fossils, which is true. However we do have quite a few examples of articulated Devonian chondrichthyans that have defied the odds and become endoskeletal fossils so in itself that doesn’t make this specimen especially interesting. Instead, I’d have thought that the cool thing about this creature is that it is a rare example of chondrichthyan endoskeleton from during the Hangenberg crisis. This is touched on in the introduction with the thorough review of the Hangenberg, but isn’t really linked to the fossil itself or to contemporaneous chondrichthyan faunas. I would encourage the authors to rejig the introduction so as to place focus on the fact that this is a Hangenberg shark, and instead of going over chondrichthyans being rare, go over what our current understanding of chondrichthyans is before, during, and after the Hangenberg. This would then be followed by the overview of the Hangenberg event itself. I think this would help the reader make more sense of the context of the fossil going into the paper.

That would also extend to the discussion: is this the only example of endoskeletal material from during the Hangenberg? If not, what else is there? Placing the jaw in its temporal context makes it more interesting.

Experimental design

This is original research within the journal's Aims & Scopes. The technical and ethical standards are high and the methods are described in detail (although see line by line comments in attached document).

The authors provide the field study permits for the work, which to the best of my knowledge are appropriate for the material described.

The research question is well defined, although I think could be reframed to make it more meaningful:

Currently the aims of the paper are framed as diagnosing the phylogenetic affinities of this jaw, with a method that the authors themselves demonstrate isn’t very good at diagnosing the phylogenetic affinities of a chondrichthyan jaw. I would recommend that instead the authors shift the focus of the paper to looking at this jaw as an example of a Hangenberg event chondrichthyan, this would keep the phylogenetic aspect, but the shape analysis would become additionally interesting as it could be used to look at shape change over the Hangenberg as well as phylogneetic affinities. I think that this would help make this paper an important contribution to the literature looking at what happened to chondrichthyans around the Hangenberg. These would only require rewriting, not further analysis, and I think could be addressed within the scope of revisions.

Validity of the findings

All data has been provided, including in supplement. Conclusions are sensible and the authors are clear about the limitations of their methods, although as above I recommend that the authors focus the article more on the Hangenberg. The anatomical work would also benefit from being a bit more systematic.

See below:

I get that it’s frustrating that this specimen has almost no phylogenetic characters with which to diagnose it, and I think that chucking it into shape analyses isn’t unreasonable as long as one is up front about the conclusions that you can draw (which the authors are). However, as the authors themselves find (and acknowledge) that there isn’t a relationship between phylogeny and shape in their data.

Instead perhaps another way of interpreting the output of the analysis is that this fossil is ecologically: it seems to be quite a generalist form of chondrichthyan? It is basically similar to everything except the weirdos (holocephalans and hybodonts). If the shape data is framed in terms of ecology then it doesn’t matter quite as much that the authors are unable to work out the phylogenetic affinities of the specimen.

Relatedly (possibly creeping beyond the scope of the paper so feel free to ignore this) something relatively quick and easy the authors could do with the data they have already collected is to plot the data on the PCO through time: is there a bottleneck in jaw shape at the Hangenberg?

To return to the taxonomic part of the paper again I realise there is a limited amount that can be made from an isolated, toothless Meckel’s cartilage. It would be futile, for example, to stick it in a phylogenetic analysis. However, I think that the attempt to diagnose it could be approached in a more systematic way. The authors correctly identify the prismatic tesselate calcified cartilage as a character allying it with more crownwards chondrichthyans . I’d recommend having a look at Brazeau & Friedman’s (2015) approach to assessing the taxonomy of isolated remains (based on the approach of others). Also maybe have a look through a recent phylogenetic analysis of chondrichthyans for characters of the lower jaw that could be applied (e.g. the dataset used for the tree in the text). Beyond that there are characters that it doesn’t have, for example it doesn’t have a terminally placed articulation, unlike holocephalans (e.g. Coates et al 2017 phylogenetic matrix), or pinched in anterior bit like acanthodiforms (see Dearden and Giles 2021). These can all be used to narrow the phylogenetic window of what it could be a little bit (even if they don’t solve it).

Additional comments

I enjoyed reading the paper and this is a cool little specimen that in my view merits publication.

As a general comment on the title and the content of the paper: the use of the word “minute”. I get that the cartilage is small from a human perspective, but is it especially small for a Late Devonian chondrichthyan? It would be quite an average size for many Devonian acanthodians or Carboniferous ‘sharks’. There is no discussion of its size relative to other chondrichthyans in the Famennian or elsewhere. So I don’t think it is really something that requires focussing on in the title (unless the authors want to discuss it in the text).

Throughout the text the words “elasmobranch” and “Holocephali” are used and it is often a bit unclear exactly what is meant. I’d recommend in all cases being specific about what is being referred to by using the total group concept: in each case are the authors referring to total-group elasmobranchs, stem-group elasmobranchs, or crown-group elasmobranchs?

Line by line comments

Line 20: ‘Chondrichthyan remains’ -> chondrichthyan fossil remains?
Line 20: ‘spines and teeth’ -> Lots of scales as well!
Line 29: ‘other chondrichthyan taxa’ -> This refers to both here and should be expanded on in the Methods: what criteria were used to choose these taxa? Approximately what breadth of time/space do they cover?
Line 30: ‘mantel -> Mantel
Line 39: ‘supposedly oldest gnathostomes’ -> This is a bit clumsily phrased, try “Gnathostomes supposedly date back to…”
Line 43: ‘Friedmann’ -> Friedman
Line 45: Notwithstanding the problems with the word ‘basal’ being undescriptive, were acanthodians ever basal on the vertebrate stem? That would put them with Myllokunmingia etc. I get what the authors mean, but maybe try framing it differently: ‘assumed primitive characters” or something.
Line 55: ‘Only teeth and fin spines’ -> Poor scales, forgotten once more…
Line 56: ‘strongly mineralised’ -> tesselate prismatic calcified cartilage is pretty heavily mineralised, maybe try ‘bony tissues’ or similar?
Line 64: “globular calcified cartilage”. -> worth making the distinction that this is a tissue that fills the whole cartilage, rather than a perichondral structure like tesselate prismatic calcified cartilage or subtessellate calcified cartilage. Also, perichondral bone has been reported in some acanthodians, such as Acanthodes, which is perhaps worth mentioning. Finally, cite Brazeau et al. 2020 here (already in refs) for synchrotron data on globular calcified cartilage in the acanthodian Diplacanthus.
Line 77ish: Should also look at (and cite) Anderson et al. 2011 “Initial radiation of jaws demonstrated stability despite faunal and environmental change” Nature
Line 80-82: Rethink the phrasing of this sentence: it reads a bit strangely
Line 84: What type of chondrichthyan teeth?
Line 104: diverse vertebrate groups -> like what?
Line 134: putting in a figure to show these stages (or referring to one in another paper) would be useful for the reader here
Line 148: It is a bit unclear from the text whether the specimen or the permit is stored in the Palaeontological institute and museum of Zurich. Define “it”.
Line 155 onwards: It would be useful to outline in a bit more detail why the methods that were chosen were chosen, over alternative metrics.
Line 156: How was it decided what taxa would be put into the jaw shape analysis? What there a taxonomic/temporal cut off point?
Line 173: Klug et al. (in prep). I realise it’s tricky juggling overlapping papers but it would be good to be able to cite at least submitted article for this given that the tree is used in the analysis: could preprint it?
Line 193: “tessellation is not as geometric as in some modern species”. I think it would be worth also explicitly comparing it to Palaeozoic taxa here. Is this lack of geometric tessellation similar to other sharky Palaeozoic sharks? Or is it “subtesselate” like acanthodians? This could be made clearer.
Line 198: “the muscle attachment” Which muscle are you inferring attaches here at the front of the mandible? Long et al (2015) figure a “muscle attachment” in this position but aren’t very clear on what they think it is…
Line 200: “we assume” why do you assume this is the tooth bearing part? Would be worth being clear.
Line 204: I think it would be worth the authors looking at the articular region in a bit more detail Is there any sign of a mandibular knob or a glenoid fossa? If so it could maybe be used as another chondrichthyan character (and should be figured). If it is not visible it should be explicitly mentioned.
Line: 207. What is a “good” separation?
Line 215: “middle and upper right side” really this describes the position in the PC1 vs PC2 plot rather than of the morphospace (which is multidimensional).
Line 220: A greater variation than what?
Line 226: Ctenacanths -> ctenacanths
Line 236: Homoplasy in what? Overall shape I guess? Probably worth being specific.
Line 262: Has anyone looked at the microfossils from this place? Are there any scales? Worth mentioning in text even if no-one has looked.
Line 280: Does the ridge extend over two thirds of the length of the jaw in the Hangenberg jaw though? Doesn’t look like it to me from the photos provided. More generally some kind of ventral ridge is in many animals, as it forms the ventral border of the mandibular adductor insertion on the mandible.
Line 293: “real tesselate” what is real tesselate cartilage? Should be defined.
Line 296 “dissolution of collagen…” This is an interesting idea: is this an original idea or is there a citation? Is there a reported example of this happening in living chondrichthyans?
Line 302: the Pers. Comm. should be reinforced by citing images of the granular texture and sections through the cartilage, which can be found in Carole’s work.
Line 317 “two depressions dorsally” this is very vague: recommend being more specific, even in summary. What features are these depressions?

Figures
The figures are good generally, and show everything in detail.
Figure 1: I think close up photos of the articular region and the symphysial region of the mandible would be benefical to the reader.
Figure 1: the caption for this figure describes it as an “ischnacanthiform acanthodian”, a conclusion not arrived at in the text…
Figure 2: The colour gradients for groups look nice, but it’s quite hard to match up the colour on the chart to the colour on the key. I’d recommend using different shapes instead (and colours for overarching groups).
Figure 3: I could have missed this (in which case, please ignore) but is Maghriboselache a published taxon? Or is that the taxon in the new phylogenetic analysis that is in prep?
Figure 3: Holocephali is spelled wrong in the key
Figure 4: Perhaps worth plotting these two values on different axes: the R statistic is uninterpretable as they are so squashed together.
Figure 5: is this most similar jaw shapes comparison based on the analysis or just done by eyeballing the shapes? Are the lines linking points of actual anatomy or just shape similarities? E.g. the region I’d interpret as probably being the articular region in the HBS jaw does not necessarily map to the articular region in the mandibles below? In both cases I think this should be made clear in the text/caption.

·

Basic reporting

The manuscript reads well, English is mostly good and professional, in some paragraphs it needs to be revised (e.g. the description on lines 192-204) to improve comprehensibility and accuracy, see annotated pdf. The introduction and background are processed sufficiently, the references are relevant. The raw data are included in the submission, although I couldn't check them because I can't open .rar files. I didn´t like how in some places the authors decide what is more or less important based on their assumptions only and not data (see lines 67-72, 156, 294-300). But the major weakness are the images. What is the purpose of Figure 3 when Figure 4 clearly shows that your analysis is unprovable. After all, it can be clearly seen that the individual phylogenetic lines are intertwined. Is this supposed to be negative evidence, that is, that morphometrics of discrete anatomical features is not a suitable method for determining the phylogenetic position of the species in question? Or did I misinterpret Figure 4? As for Figure 5 – you compare outlines, but in the description, you state that the outline of the HBS jaw is not completely preserved. What is the validity of this observation? I am asking for a more thorough explanation in the text and legends of what the individual figures are about and why they were used. It might be worth adding a picture (diagram) showing the differences between "tessellated calcified cartilage" and "globular calcified cartilage", since this is one of the main arguments of the conclusion. For more, see the annotated pdf.

Experimental design

From your analysis, the HBS jaw is closest to some acanthodians (line 224-230 and 246-249). At the end of the manuscript, however, you dismiss the results of your own analysis by claiming that the shape of the tesserae is superficially more similar to elasmobranchs, that is, you prefer comparison based on appearance, not hard data. On what basis do you consider this a more valid character than the data resulting from your analysis? I know that you explain on lines 255-261 that the method has its limits, also that it does not consider the differences in the thickness of the element (line 254-255), but only its outline. I personally have no experience with morphometric analysis, so I will not comment on the details of the methodology (which is sufficiently referenced).
This is original research, but I am not sure if the research question has been defined and what the question actually is. The analysis, carried out with technical precision and within the standards, was certainly very laborious and time-consuming. However, this fact alone should not be the reason why the research conclusions should be published. You yourself admit that the data obtained from the analysis did not move you anywhere. Therefore, I am asking you to describe and substantiate with arguments what your decision to publish these data is based on and what is the main aim of your manuscript. I am not saying that the manuscript (after incorporating the required modifications) should not be published, but as a reviewer I insist on answering the questions Why? and For what/whom? should this research be published. How exactly does your research fill the knowledge gaps?

Validity of the findings

The manuscript has two aspects; on one hand, it is a report on a unique and significant discovery, and on the other, an unsuccessful attempt to determine the element with the help of morphometric analysis. I'm not entirely sure what the aim of the manuscript is. Why did the research team decide to publish their conclusions in this particular format? If it is supposed to be a report about the unique discovery of a cartilaginous lower jaw in layers in which no cartilaginous fossils have been found, it is certainly important. I realize how rare the preservation of cartilage is in the fossil record. However, one paragraph would be enough for such a report. Or it should be a methodological article that should indicate the way how the determination of problematic, isolated and poorly preserved fossils could be solved in the future. That would certainly also be significant and necessary. Is it really an innovative way? Think about whether this aspect could be emphasized more in the manuscript.

Additional comments

The manuscript is written in good, mostly comprehensible language, definitions and background are clear. The methodology is precise, understandable, up-to-date and probably useful for future researchers. Before publication, it is necessary to work on clarifying the aims and improve the expression of ideas, especially in the discussion and conclusion parts. Then I recommend the manuscript for publication. You can find specific comments and suggestions for modifications, which I believe will improve the clarity of individual parts of the manuscript, in the annotated pdf.
Valéria Vaškaninová

---

## Round 0.2 · Minor Revisions

Dear authors,

I received the review report of Dr. Michel Laurin, who is one of the editors of the Special Issue to which you submitted this contribution.

I agree with the changes that Dr. Laurin has required, some of which had been in part already marked by the reviewers at the first round about the characters that you used to solve (provisory) the taxonomic and phylogenetic signals of the specimen. I think that by solving these minor problems the article will be ready to be included in our Special Issue.

Best regards,
Graciela Piñeiro

·

Basic reporting

This paper, which has already undergone a round of revisions, still requires minor tweaking before publication. I have annotated profusely the pdf file, which the authors will need to download, so here, I will only summarize the key points.

More background is required in some sections. For instance, for the Mantel test, we need to better know what was correlated. I am guessing phylogenetic distance and phenotypic distance, but how was the latter quantified? Similarly, paraphyly of acanthodians should be mentioned. Currently, the draft only mentions that they are stem-chondrichthyans, which is correct (given the recent literature), but this does not imply paraphyly (acanthodians could still form a clade of stem-chondrichthyans, but this is not what the papers that I have seen suggest).

I recommend replacing names of paraphyletic groups, whenever possible, by clade names. Thus, “invertebrates” can be replaced by “metazoans” or “protostomians” or “ecdysozoans”, or “lophotrochozoans”, etc.

The language still needs improvement. I have noted various typos, grammatical errors, and punctuation problems on the pdf file.

After these minor problems are fixed, I believe that this will be a useful contribution to science.

Best wishes,

Michel Laurin

Experimental design

See "Basic reporting".

Validity of the findings

Seem valid.

Additional comments

None. See "Basic reporting".

---

## Round 0.3 · accepted · Accept

Dear authors,

Considering the new modifications that you made to improve the manuscript, particularly showing the difficulties you have demonstrated exist beyond the taxonomic identification of fragmentary fossil materials, I will accept this last version as publishable for PeerJ and consider it as a very good first example representing the aim of our Special Issue. Figures are very nice and informative.
I have yet seen some very minor errors in the grammar. Please revise text construction and words at lines 130-131, 328.

Congratulations, good work!
Best wishes,
Graciela